# Swept-Source Optical Coherence Tomography-Based Biometry: A Comprehensive Overview

**Alfredo Borgia** [1,2,†], **Raffaele Raimondi** [3,†], **Tania Sorrentino** [3], **Francesco Santoru** [3], **Matilde Buzzi** [3], **Vittorio Borgia** [4], **Vincenzo Scorcia** [5] and **Giuseppe Giannaccare** [5,*]

1   Eye Unit, Humanitas-Gradenigo Hospital, 10153 Turin, Italy
2   St. Paul's Eye Unit, Department of Corneal Diseases, Royal Liverpool University Hospital, Liverpool L7 8XP, UK
3   Department of Biomedical Sciences, Humanitas University, Pieve Emanuele, 20072 Milan, Italy
4   Department of Veterinary Medicine, University of Bari, Valenzano, 70121 Bari, Italy
5   Department of Ophthalmology, University Magna Græcia of Catanzaro, 88100 Catanzaro, Italy
*   Correspondence: giuseppe.giannaccare@unicz.it
†   These authors contributed equally to this work.

**Abstract:** The purpose of this study was to summarize the results related to ocular biometry performed using swept-source optical coherence tomography (SS-OCT). A literature search was conducted to search articles reporting the clinical outcomes of patients who underwent examinations with commercially available SS-OCT machines. The available data were thoroughly analyzed, with a particular focus on all the biometric factors used to calculate the power of intraocular lenses (IOLs) implanted during cataract surgery. The agreement, repeatability, and reproducibility of several parameters among different devices were examined. The variations found for parameters obtained from agreement testing were evaluated in order to promote the interchangeability of devices. Swept-source optical coherence tomography biometers usually produce highly repeatable and reproducible results. The excellent results obtained led us to the conclusion that optical biometers based on SS-OCT technology will probably take the lead in ocular biometry.

**Keywords:** swept-source OCT; OCT; cataract surgery; ocular biometers; refractive outcomes





## 1. Introduction

Cataract surgery is the most performed surgery in ophthalmic practice, and as patients' expectations are continuously increasing, there is the need to enhance surgical outcomes. Biometers measure several parameters that are employed in various calculation formulas to predict post-operative refractive error. The two main parameters used are corneal power (K), which can be measured by keratometry or topography, and the axial length (AL), which is the distance between the corneal surface and the retina. Another important parameter in biometrical formulas is the effective lens position (ELP), which can be derived from other data such as anterior chamber depth (ACD) and lens thickness (LT). Ultrasound biometry has been the standard in clinical practice for several years; however, it is less precise in assessing AL and has now been replaced by optical methods that are faster, non-contact and more reliable. Optical methods include partial coherence interferometry (PCI), optical low-coherence reflectometry (OLCR), and swept-source optical coherence tomography (SS-OCT). "Swept source" denotes the specific type of laser used in the OCT device. Short-cavity swept lasers are used in SS-OCT in replacement of the super-luminescent diode laser typically used in standard spectral domain OCT (SD-OCT). Although the swept-source laser has a wavelength centered at approximately 1 μm, the laser sweeps across a narrow band of wavelengths with each scan [1]. Similar to SD-OCT, SS-OCT does not employ a spectrometer but has a fixed reference arm. Instead, a complementary metal oxide semiconductor camera is adopted, along with two fast parallel photodiode detectors.

Additionally, SS-OCT has a high axial resolution of 5 μm and an improved signal-to-noise ratio. The high imaging speed of SS-OCT enables the acquisition of high-resolution pictures while minimizing the unfavorable influence of patient eye movements on scan quality. In addition, SS-OCT delivers an invisible light source, which is less disturbing for patients than the visible light used in SD-OCT. Since the long wavelength of SS-OCT is less susceptible to light scattering by RPE, the imaging of deep structures is achievable. In comparison to conventional OCT, SS-OCT may be able to produce superior images in patients with cataracts because there is less light scattering by lens opacity. The purpose of this article is to summarize the current literature available on the outcomes of SS-OCT-based biometers.

## 2. Literature Search

In September 2022, a comprehensive search was performed using Embase, MEDLINE, Web of Science, and Google Scholar to ensure efficient coverage. The terms "PCI biometry", "swept source biometry", "OLCR biometry" were used and yielded 305 results dated between November 1968 and October 2022. The last research query was on 24 October 2022. The abstracts were reviewed and relevant articles were selected. The cited publications that were relevant to the subject of this review were also collected from these articles. The devices were analyzed in alphabetical order.

## 3. Swept-Source Optical Coherence Tomography (SS-OCT)-Based Biometers

This section analyzes the main technical features of four among the most popular SS-OCT biometers: ANTERION, ARGOS, IOLMaster 700, OA-2000; additionally, the repeatability and reproducibility of equipment results from different authors' use are examined.

### 3.1. ANTERION

The ANTERION (Heidelberg Engineering, Heidelberg, Germany) is an SS-OCT-based device that is capable of carrying topography, corneal tomography, anterior segment metrics, AL measurements, and IOL calculations. The device uses a 1300 nm wavelength with a scanning rate of 50,000 scans per second and 10 μm of axial resolution. The corneal anterior and posterior curvature is measured with 65 B-scans on 8 mm using a refractive index of 1.3375 for the anterior cornea and 1.376 for the posterior cornea.

Ruíz-Mesa, R. et al. published the repeatability results on ANTERION [2]. The central corneal thickness (CCT), aqueous depth (AQD), LT, AL, and pupil diameter (PD) values were presented. The within-subject standard deviation ($S_w$) for CCT was less than 1 μm, and the variability coefficient (CoV) was 0.13%, with limits of agreement (LoA) width <4 μm. With Coefficient of repeatability (CoR) < 0.15 μm and CoV < 2%, the parameters obtained for AQD and LT were comparable. Regarding AL, CoV was 0.02% (ICC = 1.000) and $S_w$ 0.004 mm. Regarding PD values, CoV and $S_w$ were, respectively, 4.06% and 0.23 mm [2]. No reproducibility findings using ANTERION have been reported up to now.

### 3.2. ARGOS

The ARGOS Advanced Optical Biometer (MOVU Inc., California, US) uses a 1060 nm wavelength and 20 nm bandwidth swept-source technology to collect two-dimensional optical coherence tomography of the full eye (SS-OCT). ARGOS measures the lengths of four segments (corneal thickness, aqueous depth, lens thickness, thickness of the vitreous humor to the retina) and then the AL is calculated as the sum of these distances. Keratometry is obtained from OCT information in combination with a 2.1 mm diameter ring made up of 16 infrared light-emitting diodes (LEDs). The unit displays the anterior corneal radius of curvature (R), the average value (RAV), and the K readings using a 1.3375 corneal index of refraction. For white-to-white (WTW) distance values, OCT image is used; however, the new version (1.5) with Alcon image guidance measures the WTW distance using the reference image.

Three studies employing ARGOS reported the repeatable results [3–5]. Nemeth et al. provided good repeatability results for K with ICC > 0.96 and CoV around 0.4% [4].

Regarding CCT, Shammas et al. identified a minimal value of Sw of 0.01 μm [3], whereas Nemeth et al. reported an ICC of 0.92–0.96 and CoV of 1.2–1.5% [4]. Additionally, for ACD $S_w$ was 0.01 and CoV 1–2%, while for LT, $S_w$ was 0.02 and CoV 0.79% [3,4]. Furthermore, Nemeth et al. reported low ICC and large CoV in ACD when pseudophakic [4]. In relation to WTW, CoV was <1% and $S_w$ was 0.11 mm [3,4].

The repeatability was remarkable for AL, for both phakic and pseudophakic eyes with ICC of 0.99 [4,5], CoV around 0.1%, and $S_w$ of 0.01–0.02 mm [3–5]. In relation to PD, ICC was 0.87 in phakic and 0.99 in pseudophakic eyes, and CoV and $S_w$ were, respectively, 5% and 0.05 mm. In addition, Shammas et al. provided reproducibility data using ARGOS; the reproducibility values were comparable to the repeatability results reported for all the parameters [3].

### 3.3. IOLMaster 700

The IOLMaster 700 (Carl Zeiss Meditec, Jena, Germany) is a biometric device that uses swept-source technology, enabling a 44 mm scan depth with 22 mm tissue resolution, to generate B-scans and to determine the biometric data of the eye measuring AL, ACD, CCT, and lens thickness [6,7]. The IOLMaster 700 provides an image of the complete eye in a longitudinal section, allowing the observer to ensure that the anteroposterior axis falls directly on the fovea [8]. The corneal curvature is measured using reflected light spots on the surface of the cornea while the pupil diameter, visual axis (line of sight), and white-to-white measurements are based on a scleral and iris image. To determine the final average K readings, three average Ks—each consisting of five single measurements—are collected. The estimated standard deviations (SD) for the ACD, LT, and AL readings are used to alert the operator to suboptimal results if the SD for ACD is greater than 0.021 mm, LT is greater than 0.038 mm, and AL is greater than 0.027 mm. Anomalous eye characteristics can be identified, such as crystalline lens tilting or decentration. Additionally, the central macular imaging obtained with IOLMaster 700 can be used as a screening strategy to detect macular disorders in patients undergoing cataract surgery [9].

Several studies on the repeatability of the IOLMaster 700 have been published over the last few years [5,6,8,10–23]. In particular, for K values the repeatability was excellent, with ICC ≥ 0.97, CoR around 0.25 D, CoV 0.22%, and a range of $S_w$ of 0.069–0.32 D [15,16]. In relation to CCT, two studies reported small Loa values and mean difference [8,12]. Additionally, the $S_w$ were comparable across studies (about 2–3 μm), the CoR < 10 μm, and a CoV < 1%; the ICC values among studies varied from 0.87 to 0.99 [8,12,15,17–22]. The WTW distance values were comparable across investigations (CoR: 0.2–0.3 mm; $S_w$: 0.10 mm), apart from the study of Shajari et al., which revealed a CoR of 0.65 mm and $S_w$ of 0.23 mm [23]. In addition, ICC was 0.87–0.99 [15,21,22], LoA between 0.32 and 0.24 mm, and CoV < 1 [8]. Regarding ACD values, there were few variations across studies, with an ICC ≥ 0.99, LoA≤ 0.06, and $S_w$ ≤ 0.05 [13]. The LT values presented $S_w$ of 0.01–0.07 [6,11], ICC ≥ 0.97, CoR up to 0.11 mm [16], and a CoV up to 1.35% [11]. The repeatability was significant for AL, with ICC of 1.000 in almost all the investigations, CoV 0.02–0.05%, CoR about 0.02, and $S_w$ ≤ 0.01.

In relation to PD, ICC was 0.87 in phakic and 0.99 in pseudophakic eyes, and CoV and $S_w$ were, respectively, 5% and 0.05 mm. The repeatability of calculating the IOL power using SRK/T and Haigis formulas was also assessed by Srivannaboon et al., who found high ICC values and minimal mean differences; there was roughly 0.60 D with respect to the LoA for both the formulas [8].

### 3.4. OA-2000

The OA-2000 (Tomey, Nagoya, Japan) is an optical biometer that combines SS-OCT with Placido disk corneal topography [24]. The SS-OCT technology, using a wavelength of 1060 nm and a scanning rate of 1250 scans/s, provides the noncontact and automatic

measurement of AL, ACD, LT, and CCT based on Fourier-domain technology that delivers high-speed tissue penetration capable of measuring through dense cataracts [25]. Moreover, OA-2000 performs several B-scans for measuring the ocular parameters to find the clearest part of the lens automatically, for measuring the highest eco and finding the path without opacity [25] However, in cases of very mature cataracts, AL and CCT measurements can also be performed with the AL-4000 handheld ultrasound biometer (Tomey), which communicates with the OA-2000 via Bluetooth technology [25]. The corneal curvature is measured using a Placido disk with nine rings, each with 256 points, projected on a 5.5 mm zone of the cornea; K values are obtained at 2.0, 2.5, and 3 mm assuming a refractive index of 1.3375 [26–29] The integrated Placido disk corneal topography allows the creation of a map of corneal shape, useful for detecting the presence of irregular astigmatism, to compare pre- and postoperative corneal shape parameters, to study eyes after corneal refractive surgery, and to identify the orientation axis for toric IOL [25]. A charge-coupled device camera captures an image thanks to an infrared light that is projected on the anterior segment of the eye and the instrument automatically scans AL, ACD, corneal diameter (CD), pupil diameter (PD), K curvature measures, CCT, LT, and topography [25].

The repeatability and reproducibility of OA-2000 were investigated by four studies [24,30–32].

In relation to keratometry, the values provided showed a good repeatability, with $S_w$ of 0.05–0.15 D [24,31], CoR of 0.15 for Km and 0.37 for steep K, a CoV around 0.20% [31], and an ICC $\geq$ 0.99. In terms of keratometry reproducibility, Wang et al. reported mean differences that were infinitesimal, with a LoA range of 0.50 D [24]. When comparing intrasession and interobserver measurements, the CoV was about 0.10%. The ICC was $\geq$0.99 [24]. Additionally, CCT values were very repeatable, with an ICC $\geq$0.98, a $S_w$ around 4 μm, CoR of 10–14 μm, and CoV < 1% [24,31]. Note that the values reported by the four investigations were comparable with respect to the CCT, the CoR value was 6–9 μm, and the CoV < 0.50%. Except for the values reported by Hua et al. [31], which were significantly larger, the WTW measurements were comparable across the investigations. Overall, the CoV was <1% and the CoR was <0.55 mm. For all the investigations, the ACD values were comparable and showed great results for each parameter. Regarding LT, the results varied among the studies, with an ICC of 0.94–0.99, CoV of 0.6–2%, and $S_w$ of 0.03–0.09 mm [24,31,32]. The AL values also showed a high repeatability, ICC was 1.000, the CoV 0.03–0.10%, the $S_w$ 0.01–0.03 mm [30–32]. The PD repeatability was only reported once, with a CoR of 1.001 mm and CoV of 5.298 [31].

## 4. Comparison and Agreement among Swept-Source OCT (SS-OCT) and Partial Coherence Interferometry (PCI) Biometers

This section describes the comparative studies among SS-OCT- and PCI-based biometers. The relevant publications on comparison between the devices and results are summarized in Table 1.

### 4.1. ANTERION vs. IOLMaster 500

Kim et al. [33] compared the ANTERION with the IOLMaster 500, reporting a good correlation and agreement; however, there was a statistically significant difference in AL, which was slightly longer in the IOLMaster 500 and in the keratometry as ANTERION measured flatter values. This can be explained by the different measuring systems. In fact, ANTERION uses 65 radial scans on a 3 mm zone, while the IOLMaster 500 measures the corneal curvature at six points in a 2.3 mm area. Nevertheless, this difference although statically significative does not carry a real clinical significance as the refractive error would minimally change. On the contrary, another study by Schiano-Lomoriello et al. did not find any statistical difference in measured parameters [34].

**Table 1.** Summary of the studies comparing Swept-Source-OCT (SS-OCT) Biometers and Partial Coherence Interferometry (PCI) Biometers.

| Author | Instrument | No. Eyes (Patients) | Main Findings |
|---|---|---|---|
| Kim et al. [33] | ANTERION, IOLMaster 500 | 175 (107) | Good correlation and agreement; flatter keratometry values in ANTERION |
| Schiano-Lomoriello et al. [34] | ANTERION, IOLMaster 500 | 96 (96) | Good correlation and agreement |
| Nemeth et al. [4] | ARGOS, Aladdin | 96 (96) | Excellent repeatability of ARGOS, except astigmatism in phakic and pseudophakic, and ACD in pseudophakic group |
| Whang et al. [35] | ARGOS, IOLMaster 500 | 153 (153) | Higher predictive accuracy of ARGOS for IOL calculations in medium-long eyes |
| Shammas et al. [3] | ARGOS, IOLMaster 500, Lenstar LS 900 | 107 (66) | AL measurements with Argos were comparable to the other biometers with a higher AL acquisition rate |
| Cummings et al. [36] | ARGOS, Lenstar LS 900 | 299 (N.A.) | The predictive accuracies of ARGOS and Lenstar LS 900 are similar, except in medium and long eyes, in which the predictive accuracy of ARGOS is higher |
| An et al. [37] | ARGOS, IOLMaster 500, Axis Nano, OM-4 | 431 (431) | No statistically significant difference in mean absolute error between ARGOS and IOLMaster 500, but the measurement failure rate was lower for ARGOS |
| Higashiyama et al. [38] | ARGOS, IOLMaster 500 | 48 (48) | The mean ALs with ARGOS were longer than those with IOLMaster 500 in the short-AL group. The mean ALs with ARGOS were shorter than those with IOLMaster 500 in the long-AL group |
| Hussaindeen et al. [39] | ARGOS, IOLMaster 500 | 376 (188) | Axial length measurements agreed among children between the ages of 11 and 17 |
| Bullimore et al. [6] | IOLMaster 700, IOLMaster 500, Lenstar LS 900 | 100 (100) | Good correlation and agreement, except for AL and mean corneal power |
| Akman et al. [7] | IOLMaster 700, IOLMaster 500 | 188 (101) | Good correlation and agreement; IOLMaster 700 more effective in eyes with posterior subcapsular and dense nuclear cataracts |
| Kunert et al. [10] | IOLMaster 700, IOLMaster 500, Lenstar LS 900 | 120 (120) | Good correlation and agreement |
| Hua Y. et al. [31] | OA-2000, IOLMaster 500 | 108 (108) | Good agreement of ocular parameters, except for the CD value |
| Reitblat, O. et al. [40] | OA-2000, IOLMaster 500, Lenstar LS 900 | 140 (90) | In 4.7% of eyes, IOLMaster 500 did not succeed in measuring AL, whereas in 94% of these cases, a reliable AL measurement was achieved with the OA-2000 |
| Vasavada, S.A. et al. [41] | OA-2000, Lenstar LS 900 | 124 (76) | Failure in AL measurements in 22.58% of eyes with dense cataract a-analyzed with Lenstar LS 900 compared with 1.6% with OA-2000. Good agreement for keratometric and ACD values. The lowest centroid error was yield by OA-2000, using the Barret toric calculator for toric IOL |

**Table 1.** *Cont.*

| Author | Instrument | No. Eyes (Patients) | Main Findings |
|---|---|---|---|
| Wang, Q. et al. [42] | OA-2000, IOLMaster 500, Lenstar LS 900 | 40 (38) | In vitreous hemorrhage, the detection rate with the OA-2000 biometer was better than that with the IOLMaster and Lenstar |
| Du, Y.L. et al. [43] | OA-2000, IOLMaster 500 | 46 (36) | OA-2000 had a lower AL measurement failure rate in myopic eyes with posterior staphyloma |
| Savini, G. et al. [44] | OA-2000, IOLMaster 500 | 249 (249) | A lower median absolute error with OA-2000 comparing its refractive outcomes to those of IOLMaster 500 in IOL power calculation |

(Devices included: ANTERION (Heidelberg Engineering, Heidelberg, Germany); IOLMaster 500 (Carl Zeiss AG, Jena, Germany); ARGOS Advanced Optical Biometer (MOVU Inc., California, US); Aladdin Biometer (TOPCON Corp, Tokyo, Japan); Lenstar LS 900 (Haag-Streit AG, Koeniz, Switzerland); Axis Nano (Quantel Medical, Cournon-d'Auvergne, France); OM-4 (TOPCON Corp, Tokyo, Japan); IOLMaster 700 (Carl Zeiss AG, Jena, Germany); OA-2000 (Tomey, Nagoya, Japan)). (ACD: Anterior Chamber Depth; IOL: Intra Ocular Lens; AL: Axial Length; CD: Corneal Diameter).

### 4.2. IOLMaster 700 vs. IOLMaster 500

Several studies showed good agreement of biometric parameters between the IOLMaster 700 and IOLMaster 500 [7,8,45]. Akman et al. compared the IOLMaster 700 and IOLMaster 500, demonstrating that the agreement between the two devices was remarkable regarding AL, ACD, K1, and K2 values [7]. In agreement, Cho et al. showed that the AL, ACD, and average keratometry values of the IOLMaster 700 could be used interchangeably with the other devices tested such as the IOLMaster 500 [46]. Huang et al. demonstrated that the success rate of axial length measurement with the IOLMaster 700 was 100% whereas the IOLMaster 500 failed in 17 out of 188 eyes [5]; conversely, various studies reported no or clinically negligible differences in AL between the IOLMaster 700 and IOLMaster 500 except for measurements in highly myopic eyes with posterior staphyloma [7,8,10]. Srivannaboon et al. evidenced that the repeatability and reproducibility of the ACD obtained by the SS-OCT device seemed better than those of the IOLMaster 500 [8]. Moreover, the IOLMaster 700 was more effective in obtaining biometric measurements in eyes with posterior subcapsular cataracts and dense nuclear cataracts. Accordingly, a statistically significant trend of higher failure rates of measurements with increasing severity of posterior subcapsular cataracts using IOLMaster 500 has been found [19,47].

### 4.3. OA-2000 vs. IOLMaster 500 in Healthy Eyes

The repeatability and reproducibility of biometric parameters measured by the OA-2000 biometer have been studied in a prospective study on 108 healthy eyes by Hua, Y. et al. that reported a high precision of this biometer, except for the pupil diameter (PD) and CD [31,48–50]. Data from the 2.5 mm zone were collected in the OA-2000 to compare corneal curvature measurement, finding that all the keratometry readings measured by the IOLMaster 500 were significantly higher than those measured by the OA-2000 [31]. However, since the differences were small (0.09 D for Kf, 0.16 D for Ks, and 0.12 D for Km), the impact of IOL power prediction was clinically insignificant [31].

### 4.4. OA-2000 vs. IOLMaster 500 vs. Lenstar LS 900 in Cataract Eyes

The longer wavelengths of the OA-2000 (1060 nm) compared with the shorter wavelengths of the IOLMaster 500 (780 nm) and Lenstar LS 900 biometer (Haag-Streit AG, Koeniz, Switzerland) (820 nm) allow a reduction in the scattering of the optic opacities for measuring the AL, ACD, LT, and CCT [40]. As is already known, the IOLMaster 500 has a limitation in its failure rate with dense cataracts and poor fixation [51]. Accordingly, the detection rate of the OA-2000 has been found to be better than the IOLMaster 500 and Lenstar LS 900 in patients with vitreous hemorrhage and very dense cataracts [42,47]. The OA-2000 has also been compared to the IOLMaster 500 and IOLMaster 700 in eyes filled

with silicone oil, resulting in it being the preferred biometer for AL measurement in these eyes [52]. In a multicenter interventional study, Savini et al. investigated the accuracy of the measurements provided by the OA-2000 for calculating the IOL power in cataract eyes [44]. They found a lower median absolute error with the OA-2000, comparing its refractive outcomes to those of the IOLMaster 500 [44]. Biometric measurements in 140 eyes that had undergone cataract extraction surgery with preoperative biometry with the OA-2000, IOLMaster 500, and Lenstar LS 900 were compared by Reitblat, O. et al., with findings of a notable advantage of the OA-2000′s success rate in AL measurement (99.7%) [40]. Indeed, in 4.7% of eyes, the IOLMaster 500 failed to measure AL whereas, in 94% of these cases, a reliable AL measurement was achieved with the OA-2000 [40]. These results agree with those of Vasavada, S.A. et al., which found a failure in axial length measurements in 22.58% of eyes with dense cataracts analyzed with the Lenstar LS 900 compared with 1.6% with the OA-2000 [41]. A high correlation was found in the average K measurements between the OA-2000, IOLMaster 500, and Lenstar LS 900 [40,41]. However, even if the mean differences were small and the agreement was high, statistically significant differences were observed for the astigmatism X and Y axes component values in the OA-2000 data [40]. The integration of the different measurements of the three instruments with different IOL power calculation formula did not show significant differences in terms of calculated target refraction errors, or for residual astigmatism in toric IOL [40]. However, since it takes into consideration the axis calculating the X and Y components of the astigmatism, the centroid residual astigmatism prediction error has been found to be the best parameter to compare refractive outcomes in astigmatic patients following the implantation of toric IOLs [53,54]. The lowest centroid error was yielded by the OA-2000, using the Barret toric calculator [40].

### 4.5. OA-2000 vs. IOLMaster 500 in Myopic Eyes with Posterior Staphyloma

The SS-OCT biometer can be very useful in cases of posterior staphyloma, an important factor affecting AL measurements [55]. Allowing the evaluation of the fixation status in myopic eyes with posterior staphyloma, the SS-OCT biometer has been described as more precise than the PCI biometer [55]. Indeed, a comparison of biometry measurements and predicted refraction in cataract patients with high myopia between the OA-2000 and IOLMaster 500 was performed by Du, Y.L. et al., showing a lower AL measurement failure rate (1.49% with OA-2000 vs. 31.34% with IOLMaster 500) in highly myopic patients (AL $\geq$ 26 mm) [43].

### 5. Comparison and Agreement among Swept-Source OCT (SS-OCT) Biometers

This section describes the comparative studies among SS-OCT. Relevant publications on comparison between the devices and results are summarized in Table 2.

### 5.1. ANTERION vs. IOLMaster 700

Fişuş et al. [56] compared the ANTERION with the IOLMaster 700, finding good agreement between the devices. Even though there was a statistically significant difference in all the parameters, the discrepancy was so small that it did not have a clinical value; however, the devices should not be considered interchangeable. Accordingly, Oh et al. [57], in a similar study, found a statistically significant difference and clinically relevant difference in the measurement of total keratometry; this was probably due to the different algorithms used to estimate this value, and may be particularly important when assessing patients that undergo refractive surgery.

**Table 2.** Summary of the studies comparing Swept-Source-OCT (SS-OCT) biometers.

| Author | Instrument | No. Eyes (Patients) | Main Findings |
|---|---|---|---|
| Fişuş et al. [56] | ANTERION, IOLMaster 700 | 389 (209) | Good correlation and agreement; minor differences in ACD and LT |
| Oh et al. [57] | ANTERION, IOLMaster 700 | 47 (29) | Good correlation and agreement, except for total keratometry |
| Moon, J.Y. et al. [58] | ANTERION, OA-2000, IOLMaster 500 | 51 (51) | Good agreement regarding AL; flatter K values with ANTERION |
| Cheng, S.M. et al. [59] | ANTERION, OA-2000, IOLMaster 700, Lenstar LS 900 | 101 (101) | Good correlation and agreement (in particular for AL data), except for CCT and IOL power prediction |
| Huang et al. [5] | IOLMaster 700, IOLMaster 500, ARGOS, OA-2000 | 171 (119) | SS-OCT biometers showed a significantly higher success rate for AL than the IOLMaster 500 |
| Yang et al. [45] | ARGOS, IOLMaster 700, IOLMaster 500 | 146 (83) | Good correlation and agreement, except for AL measurements by ARGOS; LT and CCT values were significantly different |
| Omoto et al. [60] | ARGOS, IOLMaster 700 | 106 (106) | Longer parameters of AL and CCT with IOLMaster 700. Longer ACD with ARGOS |
| Tamaoki et al. [61] | ARGOS, IOLMaster 700, OA-2000 | 622 (622) | Good comparison and agreement, except for IOL calculation in long eyes. ARGOS showed slightly myopic refraction error |
| Sabatino et al. [19] | ARGOS, IOLMaster 700 | 218 (112) | Good correlation and agreement for all parameters except for corneal diameter |
| Liao, X. et al. [62] | OA-2000, IOLMaster 700 | 103 (103) | Excellent agreement on ocular biometric measurements and astigmatism power vectors |
| Montés-Micó, R. et al. [63] | Aladdin, AL-Scan, ARGOS, IOLMaster 700, Lenstar LS 900, OA-2000 | 150 (150) | Good repeatability and agreement among devices |
| Zhang, J. et al. [52] | OA-2000, IOLMaster 700, IOLMaster 500 | 68 (68) | IOLMaster 500 and IOLMaster 700 overestimate the AL in silicone oil-filled eyes |

(Devices included: ANTERION (Heidelberg Engineering, Heidelberg, Germany); ARGOS Advanced Optical Biometer (MOVU Inc., California, US); IOLMaster 500 (Carl Zeiss AG, Jena, Germany); IOLMaster 700 (Carl Zeiss AG, Jena, Germany); Lenstar LS 900 (Haag-Streit AG, Koeniz, Switzerland); OA-2000 (Tomey, Nagoya, Japan)). (LT: Lens Thickness; K values: aximum and minimum corneal power; CCT: Central Corneal Thickness).

### 5.2. ANTERION vs. OA-2000

A recent study conducted by Moon, J.Y. et al. evaluated the level of agreement of the biometric measurements between the ANTERION, OA-2000, and IOLMaster 500 [58]. The only variable that showed high agreement and high correlation between the OA-2000 and ANTERION was AL (95% LoA $\leq$ 0.17 mm), whereas all the K values measured by ANTERION were flatter [58]. These results also justify the low agreements in the predicted IOL power among the devices [53]. Indeed, regardless of formulas, the predicted IOL powers of the three biometers were not interchangeable (95% LoA $\geq$ 1.04 D) [58]. Indeed, regardless of formulas, the predicted IOL powers of the three biometers were not interchangeable (95% LoA $\geq$ 1.04 D) [58]. Similar results were found by Cheng, S.M. et al., who analyzed biometric data provided by ANTERION, IOLMaster 700, Lenstar LS 900, and OA-2000 of 101 eyes. Except for CCT and predicted intraocular lens power, they obtained a good agreement for all parameters, and in particular for AL data, which have the best agreement between the four instruments [59].

### 5.3. ARGOS vs. IOLMaster 700

The agreement of measurements between ARGOS and IOLMaster 700 was reported in five studies, all conducted in cataract patients [5,19,45,60,61].

Three studies reported statistically significant differences in the K values measured with both devices [19,35,56]. However, being such a very small difference (0.10 D approx-

imately), it may be considered clinically insignificant. The two biometers use different methods for corneal measurements: the IOLMaster 700 obtains K values projecting light onto the cornea, while the ARGOS combines OCT and LED ring.

In relation to CCT and WTW, both devices showed statistically significant differences, with mean differences ranging from 6 to 25 microns for CCT and 0.30 mm for WTW [19,45,60]. The cause of such discrepancy may be that the IOLMaster 700 measures the corneal contour with a camera image, while the ARGOS uses the OCT image to identify the junction of the posterior cornea and iris.

Regarding ACD and LT, most of the authors reported statistically significant differences [19,45,60,61]. The AL findings varied among authors, with mean differences ranging from 0 to 0.07 mm [5,19,45,60,61]. According to Tamaoki et al., in eyes with long axial length, the refractive prediction error was slightly more myopic when using ARGOS compared to IOLMaster 700. The authors suggest this occurred because the lens constants in the Haigis formula were optimized using the measurements of axial length based on the equivalent refractive index [61]. Conversely, Omoto et al. demonstrated that the refractive outcomes using segmental refractive indices (i.e., ARGOS) instead of those utilizing the equal refractive index (i.e., IOLMaster 700) had a considerable hyperopic tendency and a reduction in arithmetic prediction errors, especially in eyes with long axial lengths [60].

In this regard, Goto et al. recently demonstrated that composite methods are less accurate than segmental methods for AL measurement with SS-OCT [64].

Finally, the AL acquisition rate was reported to be significantly higher for ARGOS than for IOLMaster 700 [61].

ARGOS may have a role in pediatric patients. Hussaindeen, J.R. et al. in their study demonstrated that AL measurements obtained with ARGOS and IOLMaster 700 were well within the clinically agreeable limits among the pediatric population, with comparable measurements for shorter and intermediate AL. The authors concluded that data from this study could be used as a reference for pediatric AL measurements, and that ARGOS could be recommended for use in the pediatric population due to its speed of acquisition and improved resolution rates [39]. Also, Higashiyama et al. demonstrated that ARGOS was useful for accurately detecting changes in the anterior segment of the eye after cycloplegia in pediatric patients. The measurements with ARGOS showed that ACD was increased and lens thickness was decreased after cycloplegia, and that ACD was increased relative to the decrease in lens thickness. No significant differences were detected in AL and CCT [65].

### 5.4. IOLMaster 700 vs. OA-2000

Comparing the biometric results between the OA-2000 and IOLMaster 700 in healthy eyes, Liao, X. et al. found a better agreement between the two biometers than with the IOLMaster 500 [62]. The mean difference for AL was 0.002 mm and for ACD was 0.004 mm ($p$ = 0.051) [62]. The reason could be found in the capability of both instruments to provide a fixation monitoring function: the OA-2000 with automatic tracing and the IOLMaster 700 with the visualization of an image along the longitudinal section to gain foveal fixation [62]. Even if the K measures obtained by the OA-2000 were significantly lower than those measured by the IOLMaster 700 ($p$ < 0.001), the mean difference values were not clinically significative in the IOL power calculation (0.04 $\pm$ 0.11 D) [62]. As regards CCT, a significative difference of 17.08 µm less in the OA-2000 measure was found, and this must be considered in the prediction formulas that include this variable [62]. A good repeatability and agreement between the OA-2000 and IOLMaster 700 on several biometric parameters was also found by Montés-Micó, R. et al. in 150 healthy eyes [63].

### 5.5. Others

To the best of our knowledge, there have not been studies conducted comparing the following devices: ARGOS vs. OA-2000, ANTERION vs. OA-2000, and ANTERION vs. ARGOS.

## 6. Discussion

This study reports an update of the literature available on a specific application of OCT technology, namely SS-OCT biometry. According to our analysis of the published literature, SS-OCT represents a promising and constantly improving technology. Comparing multiple studies, small differences, often non-statistically significant, were found among the SS-OCT devices; indeed, IOL calculation was reliable in all the considered systems (Tables 1 and 2). The SS-OCT biometers offer advantages compared to other methods of evaluating biometry, providing repeatable and reproducible measurements, even in eyes with dense cataracts. The capability of measuring AL in dense cataracts is particularly important because it avoids the use of ultrasound measurement, which is operator-dependent and can be time consuming. High volume cataract surgery clinics can particularly benefit from these fast and reliable instruments.

The SS-OCT biometer is non-invasive, and its high speed allows the collection of two- or three-dimensional data in hundredths of milliseconds with high lateral resolution and axial resolution. Among the benefits of SS-OCT are not only the high success rate of AL measurement, but also its low refractive prediction error, making it useful for accurate refractive correction.

The properties of the device in terms of repeatability and reproducibility, ease of use, integrated diagnostic tools, and the demands of the department based on the technologies already accessible must be contemplated while choosing among the various devices. In cases where the clinic lacks corneal tomography, a SS-OCT biometer such as ANTERION could be a great option, providing high-quality corneal maps and anterior segment metrics. Other SS-OCT biometers, such as the IOLMaster 700, instead allow the scanning of the macula during acquisition, which is a useful screening process for macular pathology in settings where retinal OCT is not routinely performed, also because macular alteration may impact the AL measurement and final refractive result. Additionally, since the cost of these devices may vary significantly, affordability considerations should be made.

Despite exceptional reliability in IOL power calculation, inconsistencies are present among the included comparison studies. Additional studies using multiple SS-OCT biometers together are necessary to thoroughly study all the parameters measured and evaluate their practical relevance. We suggest that further studies on the same population are required to clarify whether SS-OCT biometer measurements are interchangeable.

## 7. Conclusions

In conclusion, given the encouraging results of clinical investigations based on the novel technique discussed in this paper, SS-OCT optical biometers may offer considerable advancements over earlier ocular biometry methods, especially in high volume clinics.

**Author Contributions:** Conceptualization, A.B., R.R., V.S. and G.G.; methodology, A.B., R.R. and G.G.; writing—original draft preparation, A.B., R.R., F.S., T.S., M.B., V.B. and G.G.; writing—review and editing, A.B., R.R., V.S. and G.G.; supervision, A.B. and G.G. All authors have read and agreed to the published version of the manuscript.

**Funding:** This research received no external funding.

**Institutional Review Board Statement:** Not applicable.

**Informed Consent Statement:** Not applicable.

**Data Availability Statement:** Not applicable.

**Conflicts of Interest:** The authors declare no conflict of interest.

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
