# Peer review of "Swept-Source Optical Coherence Tomography-Based Biometry: A Comprehensive Overview"

_photonics, doi:10.3390/photonics9120951_

Round 1

Reviewer 1 Report

In this review, the authors summarized characteristics of several SS-OCT devices and compared their performances based on published articles. The selected SS-OCT devices are widely used in clinic worldwide. This review will benefit doctors who are planning to order SS-OCT and researchers who are seeking for proves to certify the repeatability and consistency as other SS-OCT devices they don’t have. 

After going through the manuscript, I am convinced that the authors have conducted detailed searching when preparing this review. I would like to suggest accepting this manuscript if the authors could add discussion of SS-OCT selection. The discussion can include the purpose of research and the best device for it. Prices of different OCTs will be also helpful.

Author Response

Thank you for the suggestion, the SS-OCT selection discussion was added accordingly. The prices and affordability of the biometers were discussed in general terms due to the large price variability of the devices mentioned in the different countries and based on the installed software modules.

Reviewer 2 Report

Cataract surgery is currently the most frequently performed surgical technique worlwide. The first step to achieve satisfactory postoperative refraction outcome is accurate ocular biometry. The authors present in their manuscript a valid comprehensive overview on optical biometry based on SS-OCT.

Minor point 

page 1 lines 34-35 please rewrite as the follow " between the cornea and the retina".

page 2 line 61 please specify literature search period performed

page 2 line 82, page 3 line 94 please check aqueous depth

Author Response

page 1 lines 34-35 please rewrite as the follow " between the cornea and the retina"

Response 1: thank you for the suggestion, that was modified accordingly

page 2 line 61 please specify literature search period performed

Response 1: thank you, the literature search period was specified

page 2 line 82, page 3 line 94 please check aqueous depth

Response 1: thank you for the suggestion, those were checked and modified accordingly